# Leveraging the Dynamic Immune Environment Triad in Patients with Breast Cancer: Tumour, Lymph Node, and Peripheral Blood

**DOI:** 10.3390/cancers14184505

**Published:** 2022-09-17

**Authors:** Isobelle Wall, Victoire Boulat, Aekta Shah, Kim R. M. Blenman, Yin Wu, Elena Alberts, Dinis Pedro Calado, Roberto Salgado, Anita Grigoriadis

**Affiliations:** 1Cancer Bioinformatics, School of Cancer & Pharmaceutical Sciences, King’s College London, Guy’s Hospital, London SE1 9RT, UK; 2Immunity and Cancer Laboratory, The Francis Crick Institute, London NW1 1AT, UK; 3Department of Pathology, Tata Memorial Centre, Homi Bhabha National Institute, Mumbai 400012, India; 4Department of Internal Medicine, Section of Medical Oncology, Yale School of Medicine, Yale University, New Haven, CT 06510, USA; 5Department of Computer Science, School of Engineering and Applied Science, Yale University, New Haven, CT 06511, USA; 6Breast Cancer Now Research Unit, School of Cancer & Pharmaceutical Sciences, King’s College London, Guy’s Hospital, London SE1 9RT, UK; 7Peter Gorer Department of Immunobiology, School of Immunology & Microbial Sciences, King’s College London, London SE1 9RT, UK; 8Centre for Inflammation Biology and Cancer Immunology, School of Immunology & Microbial Sciences, King’s College London, London SE1 9RT, UK; 9Department of Pathology, GZA-ZNA Hospitals, 2610 Antwerp, Belgium; 10Division of Research, Peter MacCallum Cancer Centre, Melbourne, VIC 3000, Australia

**Keywords:** systemic immunity, breast cancer, lymph node, triple-negative breast cancers

## Abstract

**Simple Summary:**

The primary, secondary and tertiary immune sites, namely the tumour microenvironment (TME), the lymph nodes (LNs) and the peripheral blood, form the cardinal trinity of immune environments in breast cancer. With the success of immuno-therapies in some subtypes of breast cancers, an integrated understanding of these intertwined immune sites is essential to potentiate their anti-cancer responses and enhance the efficacies of therapeutic agents, and in turn, illuminate novel pathways of anti-cancer immunity and therapeutic opportunities.

**Abstract:**

During the anti-tumour response to breast cancer, the primary tumour, the peripheral blood, and the lymph nodes each play unique roles. Immunological features at each site reveal evidence of continuous immune cross-talk between them before, during and after treatment. As such, immune responses to breast cancer are found to be highly dynamic and truly systemic, integrating three distinct immune sites, complex cell-migration highways, as well as the temporal dimension of disease progression and treatment. In this review, we provide a connective summary of the dynamic immune environment triad of breast cancer. It is critical that future studies seek to establish dynamic immune profiles, constituting multiple sites, that capture the systemic immune response to breast cancer and define patient-selection parameters resulting in more significant overall responses and survival rates for breast cancer patients.

## 1. Introduction

In 2020, breast cancer became the most commonly diagnosed malignancy worldwide and remained the leading cause of death amongst women [1]. Multi-faceted treatment regimens which combine two or more of surgery, chemotherapy, radiation and targeted therapies have significantly improved outcomes for breast cancer patients. Immune checkpoint blockade (ICB) with pembrolizumab and atezolizumab has improved progression-free survival (PFS) and overall survival (OS), respectively, particularly in patients with Triple-Negative Breast Cancer (TNBC) [2,3]. Nevertheless, overall response rates (ORR) are variable, and patient-selection biomarkers show suboptimal sensitivity and specificity. Initial efforts have focused on exploring the dynamic tumour-microenvironment (TME). Still, the frequency and severity of adverse systemic effects for the majority of these treatment regimens indicate a broader involvement of immune sites in the anti-cancer response. Indeed, **systemic** immune features are shown to have prognostic significance in breast cancer and are critical in the elimination of cancer cells [4].

The primary, secondary and tertiary immune sites, namely the TME, the lymph nodes (LNs) and the peripheral blood, form a dynamic immune *macro*environment that shapes the net immune response to cancer (Figure 1) [5]. These three sites not only host, traffic and activate essential immune cell populations, but they also exhibit an array of anti-tumour activation pathways and facilitate the seeding of cancer cells. High resolution multiplex technologies, including single-cell RNA sequencing [6], mass cytometry [7], spatial transcriptomics [8] and computational-based analyses of histological specimens, have provided a wealth of spatial, cellular and phenotypic immune profiles of the TME and circulating leukocytes in breast cancers [9]. These investigations have revealed predictive biomarkers for response to treatment in both the neoadjuvant and the adjuvant settings [9,10]. Despite being the first site of cancer cell seeding, the LN has so far obtained less attention beyond the detection of metastatic deposits. Whilst we and others have reported on the prognostic value of morphological changes in cancer-free LNs, this site is often overlooked. In the majority of past studies, a static view of wider immune responses to breast cancer is presented, disregarding the **dynamic immune cross-talk** that occurs between these areas [11]. Here, we discuss how immune responses at the primary tumour site, the LN and the peripheral blood indicate an intrinsic and continuous communication in breast cancers, with a focus on TNBC. We review how the presence of tumour and disease progression can both suppress and activate immunity at multiple sites (Table 1), how current therapies show responses across the immune *macro*environment with time-associated profiles and highlight the governmental role of the LNs in systemic immune responses to breast cancer. Thereafter, we focus on how combination therapies have been used to optimise responses to immunotherapy, both locally and systemically.

## 2. The Tumour Microenvironment, Immune Features beyond Tumour Infiltrating Lymphocytes

In addition to quantitative analysis of tumour infiltrating lymphocytes (TILs), breast carcinomas can also be categorised based on the spatial organisation of their TILs. To date, four spatial-immune phenotypes have been described: (i) *immune-excluded*, in which immune cells are present in the tumour margins but cannot penetrate cancerous tissue; (ii) *immune-desert*, with an absence of immune cell infiltration; (iii) *inflamed—stromal and intratumoural*, also known as ‘dispersed’, in which immune cells infiltrate throughout the tumour bed and stroma but cannot form organised networks; and (iv) *inflamed—stroma restricted*, when immune cells are found exclusively in the stroma and often form aggregates [12,13,14]. PD-1+ CD4+ T cells and PD-L1+ tumour cells are associated with *inflamed—stroma restricted* subtypes, whilst PD-L1+ CD8+ T cells and PD-1+ tumour cells dominate in the *inflamed—stromal and intratumoural* subtype [13]. Different spatial-immune phenotypes in the TME have been associated with survival outcomes, even amongst patients with corresponding cellular meta-clusters from the same clinically-defined cohort [58]. Patients with an *inflamed* TNBC subtype demonstrate significantly improved metastasis-free survival (MFS), disease-free survival (DFS) and OS compared to those with *immune-desert* or *immune-excluded* phenotypes [59]. Nevertheless, spatial-immune profiling of histopathological slides seems to be poorly reproducible between pathologists, and high variability exists in both an inter- and intra-tumoural manner [9]. Digital pathology combined with deep learning methodologies have further highlighted the complexity of TILs distribution in TNBC and, in the future, may provide a deeper characterisation of spatial heterogeneity within the tissue.

Immune cell aggregates, such as tertiary lymphoid structures (TLSs), have sparked great interest due to their relevance to immunotherapy responses in multiple cancer types [60]. TLSs are highly organised ectopic lymphoid organs, with a central B cell zone surrounded by a T cell zone and a collar of antibody-producing plasma cells. Interspersed throughout the TLSs are high endothelial venules (HEVs) and distinct populations of dendritic cells (DC), such as CD21+ follicular DCs (FDC), which are essential to the active germinal centre responses observed in matured B cell follicles of both TLSs and the LNs [60]. The specific interplay between TILs within TLSs remains unclear [61]; nevertheless, their strong structural resemblance to the LN suggests an intimate relationship between the immune responses at both sites (Figure 2). In breast cancer, a 12-chemokine signature capable of predicting the presence of TLSs is shown to be prognostic for improved survival [15]. Concurrently, the presence of histologically-detected TLSs is prognostic for OS in TNBC [16] and related to improved DFS and OS across multiple breast cancer subtypes [17,62]. TLSs are found in different stages of maturity [63] and are typically transient. Whilst the identification of mature TLSs is readily feasible for standard pathology laboratories, diagnostic pathological sectioning may fail to detect the presence of immature or non-persistent TLSs. Thus, standards to report on their presence are urgently needed to truly establish their clinical relevance for breast cancer.

The breast TME contains a multitude of immune subsets, which hold established prognostic value independently of their spatial organisation. In short, anti-tumour effector cells, including CD4+ helper T cells and CD8+ cytotoxic T cells, in addition to the ratio of CD8+ cytotoxic to FOXP3+ T regulatory (Tregs), are associated with better patient prognosis and response [64]. On the other hand, immunosuppressive cells such as myeloid-derived suppressor cells (MDSCs), tumour-associated macrophages (TAMs), Tregs and neutrophils are abundant in the TME and associated with a worse outcome [65]. Deep spatial, cellular and molecular characterisation of TILs has recently shed light on two immune cell types with intriguing roles in anti-tumour immunity, B cells and γδ-T cells. B cell populations largely correlate with improved outcomes in breast cancer patients, particularly those in TNBC [66,67]. Tumour-infiltrating B cells express activation molecules, produce cytokines and antibodies, and are responsive to B cell receptor (BCR) stimulation when isolated [18]. In TNBC, a highly activated IgG isotype-switched tumour-infiltrating B cell population is associated with a favourable prognosis [19]. A more precise characterisation of B cell subsets to understand their multilateral influences on tumour development, anti-tumour responses and influence on treatment, particularly regarding the production of auto-reactive antibodies, remains to be explored.

The γδ-T cells, a sparse population in solid tumours, lie at the intersection of innate and adaptive immunity and are capable of both suppressing and promoting tumour growth (reviewed in [68]). Unlike conventional αβ-T cells, their T cell receptor (TCR) is composed of γ and δ chains. They mostly lack expression of the CD4 and CD8 co-receptors and do not recognise antigens in an MHC-dependent manner [39,69]. Instead, they recognise cell stress ligands on transformed cells via innate receptor molecules such as NKG2D and DNAM-1. They have multiple functions including both direct and indirect antibody-dependent cellular cytotoxicity (ADCC), as well as providing antigen presentation help to conventional CD4+ and CD8+ T cells. Both healthy breast and paired TNBC tissue harbour resident innate-like Vd1+ γδ-T cells, which can sense transformation via NKG2D and enable tumouricidal Th1 and cytolytic responses [20], and their prevalence is positively associated with PFS and OS in TNBC [20]. Despite these anti-tumour properties, an immunosuppressive role for a subset of IL-17-producing γδ-T cells in a murine model of breast cancer has been established, [38]. In this model, IL-17-producing γδ-T cells express IL-10, IL-8 and CD73, dampen naïve and effector T cell responses and proliferation, and block DC maturation and function [21,22]. Nonetheless, IL-17-producing γδ-T cells have been exceptionally hard to find in humans [20,40] and their potential to mount immune responses, independent of MHC-activation, make γδ-T cells attractive targets for “off-the-shelf” immunotherapy for breast and other cancers.

## 3. Peripheral Immune Responses, a Cross-Talk with the Primary Tumour

Synergistic to direct anti-tumour immune responses at the TME, the peripheral blood and the LNs represent reservoirs and activation sites of immune cells during cancer progression [10], exemplifying their intimate communication with the primary tumour. Tumour-induced hematopoietic dysregulation is common in multiple cancer types, including breast cancer [5], whereby the overall proportion of immature myeloid cells in the peripheral blood [5], and the frequencies of circulating granulocyte–monocyte progenitors increases [41]. A high neutrophil to lymphocyte ratio is associated with a worse prognosis in multiple breast cancer subtypes [23,24], and a high lymphocyte to monocyte ratio is predictive of an improved response to treatment [23]. High circulating lymphocyte counts are consistently associated with better outcomes [25]. Intriguingly, increased TIL levels (>50%) at the primary tumour do not correlate with increased absolute circulating lymphocyte counts in breast cancer patients and are negatively associated with absolute circulating neutrophil counts [42]. However, cluster analyses of RNA expression profiles of peripheral blood and patient-matched tumours revealed an association between peripheral inflammation and intra-tumoural immune activation [26].

Compared to healthy donors, the peripheral blood of breast cancer patients harbours functionally impaired mature T cell subsets with reduced secretion of pro-inflammatory cytokines (IL-2, IFN-γ), defective responsiveness to IL-6, reduced T cell receptor (TCR) signaling [27], and circulating CD8+ T cells display senescent (KLRG1+NKG2+) or exhausted (CD27−CD28−PD-1+) phenotypes [28]. In TNBC patients, the frequency of terminally exhausted CD4+ T cells and CD8+ T cells, characterised by the expression of CD39, is highly correlated between the primary tumour and the peripheral blood [29], potentially reflecting an intimate relationship between T cell suppression at both sites. Tumour-induced circulating NK cells express decreased activation receptors [30] and are less cytotoxic, exhibiting a reduced capacity for direct killing of cancer cells and secreting immunomodulatory cytokines.

As APCs, DCs play a crucial role in mounting systemic immune responses. Compared to their healthy counterparts, circulating DCs display fewer mature phenotypes, lower levels of the activation markers CD40, CD86, HLA-DR and CD83 [31], and produce less TNF-α, in stimulatory conditions [32]. In preclinical models of breast cancer, tumour-induced G-CSF suppresses the differentiation of circulating pre-DCs into mature DCs [33], leading to a reduced number of antigen-specific CD8^+^ T cells and reduced migratory DC numbers to the tumour-draining LNs [33].

As well as dysfunctional immune cells, the peripheral blood of breast cancer patients harbours immunosuppressive cells, many of which are also found at the primary tumour. For instance, breast cancer patients exhibit greater numbers of circulating FOXP3+ T cells compared to healthy volunteers [43], a particularly immunosuppressive subpopulation of FOXP3^hi^/CD45RA-Tregs which closely resembles an intratumoural, both in their expression of CD39, CTLA-4, TIGIT and CCR8 and their clonal diversity [34]. Though the number of peripheral Tregs in breast cancer patients is not prognostic, patients in whom Tregs are more responsive to immunosuppressive cytokines exhibited shorter recurrence-free survival (RFS) [34]. In breast cancer patients, peripheral blood monocytes, progenitors of TAMs which differentiate upon recruitment into tumours, are less responsive to IFN-γ and IFN-β [35]. Interestingly, IFN-γ responsiveness in these peripheral monocytes is negatively associated with the degree of infiltration by TAMs in patient-matched breast tumours, suggesting that impaired IFN-γ signaling in blood monocytes may drive the recruitment of immunosuppressive TAMs to the primary tumour [35]. MDSCs are found to be increased in the blood of breast cancer patients, and their abundance correlates with disease stage, residual status [36] and high Tregs frequencies [37]. Peripheral MDSCs exhibit typical markers of suppression, including downregulated CD80/86, TNF-α, IL-1β, matrix metalloproteinase and arginase [37]; they also inhibit T cell proliferation [37,70]. In murine breast cancer models, G-CSF-elicited expansion of circulating neutrophils leads to metastasis through the suppression of CD8^+^ T cell proliferation and effector function [38]. Increased circulating G-CSF stems from IL-17 production by tumour-infiltrating γδ-T cells [38], which in turn depends on CCL2-mediated production of IL-1β by TAMs [71]. The system-wide immunosuppressive role of γδ-T cells in this model further demonstrates their functional ambiguity in anti-tumour immune responses.

In addition to leukocytes, platelets also contribute to the immune environment of the peripheral blood. The role of platelets in haemostasis and wound-healing is well established; however, they facilitate several immune-related functions [72]. Platelets can express Toll-like receptors (TLRs) allowing their binding to pathogens [73], they secrete a plethora of inflammatory mediators [74] and may act as APCs for CD8+ T cells [75]. Critically, they can facilitate immune evasion of tumour cells. Platelets are well known for their role in promoting cancer progression and metastasis via the ‘cloaking’ of tumour cells, promoting MICA and MICB shedding and subsequently shielding them from NK cell-mediated elimination [76], as well as secreting TGF-β [77] and other immunosuppressive mediators which promote metastasis. Cancer patients are found to harbour an increased platelet to lymphocyte ratio compared to healthy donors [78]. As is also observed in multiple cancer types, a low circulating platelet to lymphocyte ratio in breast cancer patients is associated with improved prognosis [55,79] and is found to be an independent prognostic factor for RFS and OS in early breast cancer [80]. A Systemic Inflammatory Index (SII), calculated as *neutrophil x platelet/neutrophil count*, is also predictive for DFS and OS in breast cancer [81] and was found to be more reliable in human epidermal growth factor receptor-2 positive (HER2+) breast cancer patients compared to other inflammatory markers [82].

Thus, the peripheral cellular immunome across all lineages is deeply restructured in response to tumour development; immunosuppressive cells are enriched, and anti-tumour effector cells are defective. This occurs in the absence of direct contact with tumour-infiltrated sites. Indeed, once monocytes leave the blood and infiltrate tissues, they remain within these sites. Hence, their peripheral dysfunction must be in response to cancer-induced effects at distant sites [35]. Intriguingly, tumour-induced haematopoietic dysfunction is reversed upon patient recovery and upon tumour resection in mouse models of breast cancer [83], highlighting the plasticity of the immune macroenvironment [84]. This provides a rationale to derepress affected sites like the periphery via targeted therapies. Indeed, in the aforementioned mouse model of breast cancer exhibiting G-CSF-dependent neutrophil expansion, the depletion of circulating neutrophils or intratumoural γδ-T cells significantly decreases local and distant metastases [38]. Additionally, hyporeactive DCs isolated from breast cancer patients and stimulated ex vivo via CD40L-conditioning increase their expression of CD86 and HLA-DR, robustly secrete IL-12 and exhibit an enhanced ability to stimulate healthy T cells to proliferate and produce IFN-γ [85].

## 4. Lymph Nodes: The Immune Capital of Anti-Tumour Responses in Breast Cancer

In the war waged between a patient’s immune system and breast cancer, the primary tumour represents the battlefield. Here, infiltrating effector immune cells combat cancer cells directly in a cancer-hijacked TME and inform on the local immune responses. On the other hand, the periphery is subject to distant immunosuppressive signals and provides a snapshot of cancer’s broad influence on the mounted systemic immune response. Meanwhile, the sentinel LNs, the first LNs to drain cancer cells, possess a double-edged relationship with the primary tumour, acting as both the first site of metastasis and of highly specific, adaptive anti-tumour responses. LNs are, therefore, a gateway to gaining a wider perspective on a patient’s ability to mount successfully anti-tumour immunity.

In breast cancer, the metastatic LNs are altered structurally, molecularly and cellularly by the presence of tumour cells. Immunologically, metastatic LNs exhibit decreased CD4+ to CD8+ T cell ratios [44] and reduced frequencies of CD1a+ DCs. This, coupled with an expansion of Tregs [46], MDSCs [45], and increased levels of the IL-10, FOXP3, CTLA-4 and PD-1 suggests a highly immunosuppressive environment [47,48]. Compared to non-metastatic LNs, Tregs in metastatic LNs express higher levels of the inhibitory molecules GITR, OX40 and CTLA-4, denoting high immunosuppressive capacity [46]. This is consistent with preclinical models, which demonstrate that elevated levels of nodal Tregs increase tumour growth and spontaneous metastasis to distant organs via TGF-β secretion [86]. Metastatic LNs exhibit an increased frequency of “exhausted” T cells, characterised by expression of CTLA-4, PD-1 [45] and TIGIT with suppressed TCR signaling [87]. Upon ex vivo stimulation, these T cells release lower levels of the inflammatory cytokines IL-4 IL-6, IL-10, TNF-α, IFN-γ and IL-17 compared to those from cancer-free LNs [45]. This phenotype of metastatic LN T cells is preceded by the suppression of LN-resident DCs [45], which are less mature [88,89] and express decreased levels of the activation markers CD40, CD86 and CD83 than healthy and cancer-free LN DCs. Upon ex vivo stimulation, DCs isolated from metastatic LNs release lower levels of IL-8, IL-1β, IL-6, IL-10, and TNF-α, compared to their healthy LN counterparts [45], suggesting that dysfunctional antigen presentation machinery leads to sub-optimal T cell priming. In parallel, metastatic LNs harbour elevated levels of M2-phenotype TAMs, thereby promoting immune tolerance via the tumour-induced secretion of indoleamine-pyrrole 2,3-dioxygenase (IDO) [49], and potentially aiding their persistence and further dissemination to distant seeding sites in the body.

The suppression of the immune responses in the LNs of breast cancer patients precedes metastasis. Genomic studies performed at different stages of LN metastasis revealed that gene patterns reflecting DC deficiencies and hyper-proliferative B cells, accompanied by a plethora of tumour-promoting pathways, significantly contribute to the transition of a pro-metastatic niche into a metastatically involved LN [90]. Cancer-free LNs harbour fewer CD4+ T cells, CD8+ T cells [44], CD80+, CD86+ and CD40+ immune cells [47] and DCs which do not show characteristics associated with effective antigen presentation [51]. On the other hand, compared to metastatic LNs, the cancer-free LN exhibits greater numbers of CD3+ T cells [52], a higher abundance [44] and more mature phenotype of DCs [51], more IL-12-secreting cells [89] and more activated NK cells subsets [91]. Thus, there is a fine line between tumour-induced immune activation and immunosuppression in the LN, as exemplified by the ex vivo secretion of inflammatory cytokines by cancer-free LN DCs and T cells, which is decreased compared to healthy controls but to a lesser extent than those from the metastatic LN [45]. Already in 1998, Wong et al. reported differences in the capacity of immune cells from the tumour, axillary LN, and peripheral blood of breast cancer patients to function normally. In matched samples, axillary LNs harboured less IFN-γ-positive T cells than the peripheral blood and the tumour. Moreover, peripheral blood leukocytes exhibited higher cytolytic activity against a breast cancer cell line than both LN and tumour leukocytes. Finally, tumour-infiltrating leukocytes from LN-positive patients showed higher tumouricidal activity than those from LN-negative patients [53].

Active immune response in LNs may provide additive value for the risk prediction of disease progression in breast cancer patients. Increased numbers of T and B cells in sentinel LNs [52] have been associated with longer DFS across different breast cancer subtypes irrespective of their nodal status, as are increased levels of axillary node CD4+ T cells and DC populations in LN-positive breast cancer patients [44,52]. Patients with enlarged LNs without metastatic involvement (indicative of an active immune response) experience longer breast-cancer-specific-survival [56]. Topically, swollen LNs are an unanticipated side effect of the immune reaction engendered by the COVID-19 vaccine which has been complicating breast cancer screenings. These enlarged LNs can appear on mammograms, leading to potential follow-up assessments [92]. The LN is also the site of the germinal centre response that facilitates the generation of affinity-matured long-lived memory B cells and antibody-producing plasma cells. The importance of the germinal centre response for long-term immunity following infection and vaccination has long been understood, including responses to SARS-CoV-2 [93]; however, it is acutely understudied in the context of cancer. The germinal centre response requires the intimate interaction of germinal centre B cells with LN-resident T follicular helper (Tfh) cells and FDCs. In the LNs, morphological substructures, including germinal centres, are altered in response to a nearby developing cancer. We have shown in two independent studies that the formation of germinal centres in cancer-free LNs is predictive for a lower risk of developing distant-metastasis in LN-positive TNBC patients [50,54]. TNBC patients with stromal TIL levels higher than 20% have more germinal centres in their cancer-free LNs, while the identification of TLSs in the TME correlates with an increased number of germinal centres in metastatic LNs [50]. In alignment with this, Quintana et al. studied similar features in a LN-negative TNBC cohort and found that TNBC patients with >50% TILs have more and larger germinal centres in their LNs and more TLSs at their primary tumour site compared to patients with <5% TILs [57]. We, too, have evidenced immune activation via the formation of germinal centres in LNs in level 1 of the axilla (closest to the primary tumour), as compared to level 3 LNs (most distal to the primary carcinoma) which do not show any germinal centre formation [54]. A cross-talk between germinal centre responses in LNs and a prognostic immune response at the primary tumour must be considered, especially given the cellular and spatial similarities between TLSs and germinal centres (Figure 2).

A functional anti-tumour role for the LN has been posited. In murine models, the exhausted T cell compartment in the TME can be replenished with a pool of stem-like Tcf7+ effector CD8+ T cells from the tumour-draining LN [94]. In humans, breast cancer patients with low TIL levels (<5%) at their primary tumour express higher levels of immune checkpoint molecules, including CTLA-4 and OX-40, in their tumour-draining LNs than patients with high TIL levels (>50%) [57], leading to the conjecture that the tumour-draining LN is one source of TILs and that T cell deactivation in the tumour-draining LN may contribute to a dampened immune response at the primary tumour site. TCR sequencing of the primary tumour and tumour-draining LN T cells revealed extensive clonotype overlap between expanded CD8+ T cells from both sites. All expanded tumour-draining LN clones were also found in the tumour, suggesting T cell priming may occur in the tumour-draining LN before tumour infiltration [95]. Moreover, T cells engineered to express TCRs from expanded clonotypes can exert anti-tumour reactivity against autologous cancer cells (though this does not apply to those from non-expanded clonotypes) [95]. The accumulating evidence, both in preclinical models and through prognostic patient biomarker studies, has resulted in a shift in our perception of LNs in cancer.

## 5. Immunomodulatory Effects of Chemotherapy throughout the Immune Macroenvironment

For certain breast cancer subtypes, such as TNBC, neoadjuvant chemotherapy (NACT) is increasingly being used as first-line treatment for patients, given its advantages in de-escalating axillary dissection, prognostication and informing subsequent treatment regimens [96]. In addition to its potent cytotoxicity against cancer and other highly proliferative cells, chemotherapy can induce immunogenic tumour-cell death, and may, thus, potentiate anti-cancer immune responses [97]. NACT frequently causes a depletion in the total number of TILs which is associated with pCR (pathologic complete response) in multiple breast cancer subtypes [98,99]. Accordingly, high or increased levels of total TIL counts post-NACT compared to baseline levels have been associated with residual disease or poor outcomes in patients with inflammatory TNBC [100]. However, further investigation is required to fully delineate the clinical significance of post-NACT TIL counts due to conflicting and variable results [17,101,102,103]. Across all breast cancer subtypes, intratumoural CD8+ T cells are increased after NACT; meanwhile, the total CD3+ T cell compartment and the proportions of CD3+ CD4+ T cells and CD20+ B cells are reduced [104]. Notably, increased CD8:FOXP3 ratios following NACT are associated with pCR in HER2+ breast cancer patients and even with RFS in TNBC patients with residual disease [105,106]. Improved therapeutic effects are also associated with increased levels of intratumoural NK cells and IL-6 across multiple breast cancer subtypes [103]. By contrast, when the effects of NACT are evaluated in estrogen receptor (ER)/progesterone receptor (PR)-positive breast cancer patients only, CD8+TILs are depleted post-NACT [107]. In addition to breast cancer subtype, the spatial localisation of TILs may also influence their response to NACT. In the intratumoural compartment, CD8+ and CD4+ TILs were significantly increased post-NACT, in contrast to those in the stromal compartment which were significantly decreased [108].

Absolute numbers of B cells post-NACT are significantly decreased, whilst their overall effector functions are enhanced, and their presence in the adjuvant setting correlates with response to NACT in breast cancer patients [109]. Additionally, TLS formation has been linked to NACT treatment in breast cancer patients, especially those who experience a therapeutic response [109]. In TNBC particularly, the proportion of a distinct B cell subtype that upregulates the inducible T cell co-stimulator ligand (ICOSL) is dramatically increased following NACT [109]. ICOSL+ B cells are found to accumulate predominantly within TLSs, accounting for 45% of CD19+ B cells after chemotherapy compared to less than 1% in the neoadjuvant setting [109]. ICOSL is a co-stimulatory molecule expressed by antigen-presenting cells, including B cells, which promotes the proliferation and differentiation of activated T cells. In the LN, ICOSL expression on B cells promotes CD40L expression on T cells and vice versa, resulting in an ‘entangled’ mode of B–T cell interactions [110]. In this way, ICOSL is critical for the survival and maturation of germinal centre B cells, an effective germinal centre reaction and the generation of long-lived memory and plasma cells [111]. Thus, chemotherapy-induced expansion of ICOSL+ B cells that localise to the TLSs, may indicate a germinal centre-like response in the TME, exemplifying similarities between immune responses at the TME and the LN.

In breast cancer patients, the effects of chemotherapy on B cells in the peripheral blood mirror those observed at the primary tumour, unlike other immune cell phenotypes. Neutropenia is a commonly reported side effect amongst breast cancer patients receiving multiple chemotherapy regimens [112]; however, limited studies have reported on the prognostic significance of the neutrophil to lymphocyte ratio post-NACT and the results are variable [113,114,115]. Meanwhile, circulating lymphocyte counts are depleted and CD8+ T cells, CD4+ T cells and NK cells are consistently reduced in representation and functionally impaired, exhibiting increased expression of inhibitory receptors (CD85, LIAR and NKG2A) and decreased expression of activator receptors (NKp46 and DNAM-1) post-NACT [116,117]. Nevertheless, some activating receptors (NKp30 and NKp44) are increased. The effects of chemotherapy on circulating B cells are reminiscent of those observed in the TME. Total B cell numbers are more depleted than other lymphocytes, but within the CD19+ B cell compartment, proportions of both CD19+ ICOSL+ B cells and CD19+CD20- plasmablasts are expanded, as observed in the TME of TNBC patients following chemotherapy [109,117]. Interestingly, most immune cell populations are seen to replenish to some degree, in contrast to B cells which do not recover even nine months post-treatment [118]. As observed with lymphocytes, platelets in the periphery are variably affected by NACT; however, their prognostic significance is not lost. When evaluating the platelet to lymphocyte ratio from breast cancer patients before and after NACT, patients with a consistently high platelet to lymphocyte ratio experienced significantly worse MFS compared to those patients with persistently low or altered PLRs [113].

The known effects of chemotherapy on individual lymphocyte populations of the LNs are so far limited. To our knowledge, only one study has reported the chemotherapy-induced effects on lymphocyte subsets in breast cancer patient LNs. In breast cancer patients of all subtypes, those who received NACT demonstrated a higher degree of lymphocyte depletion and were more likely to experience LN lymphopenia compared to those treated with neoadjuvant surgery [119]. Chemotherapy was more likely to result in LNs with reduced B cell zones compared to surgery; meanwhile, T cell zones were less affected in both groups. These results broadly mirror what is seen in both the TME and the peripheral blood following NACT (Figure 3) but reflect a critically unexplored field.

## 6. On-Treatment Assessment of Immune Responses

In contrast to post-NACT, where TILs are markedly reduced compared to baseline levels across all breast cancer subtypes, short-term chemotherapy increases the number of absolute TILs, enhances cytotoxic and inflammatory pathways and upregulates gene-expression signatures predictive of response to anti-PD-1 therapies [120,121,122]. Between on-treatment and post-NACT analyses, fractions of CD4+ T cells and M1-macrophages were reduced in patients who achieved pCR, while those with residual disease exhibited increased proportions of the immunosuppressive mast-cells and M2-TAMs. Immune induction with doxorubicin led to an ORR of 35% compared to 20% across the entire cohort [121], whilst increased TILs counts and CD8+ T cell fractions at ‘on-treatment’ analysis were predictive for pCR to NACT across all breast cancer subtypes. ‘On-treatment’ analysis of immune features correlated strongly with outcome compared to baseline or post-treatment levels [120]. ORRs to ICBs increased to 35% in patients with metastatic TNBC who received induction chemotherapy compared to 17% in those who did not receive induction therapy [121]. Indeed, inflamed spatial phenotypes were enriched in TNBC patients who exhibited a clinical response to short-term chemotherapy followed by anti-PD-1 treatment [59]. However, *immune-excluded* and *immune-desert* spatial phenotypes were enriched in TNBC patients who did not respond, illustrating that the multi-faceted influence of chemotherapy may be affected by other factors. In addition, ‘on-treatment’ TILs evaluated 15 days after the administration of trastuzumab were found to be a better predictor of pCR in HER2+ breast cancer compared to pre-treatment values [123]. Together, these findings indicate that transient chemotherapy induces an enhanced immunological landscape in breast carcinomas that are distinct from those observed at baseline or post-NACT.

The assessment of ‘on-treatment’ responses at immunological sites other than the primary tumour has so far been sparse. After three weeks of chemotherapy treatment, the peripheral blood of HER2+ breast cancer patients displayed increased proportions of CD3+ T cells and CD3+ CD8+ T cells, in contrast to reduced levels of CD3+ CD4+ T cells, CD19+ B cells and NK cells [124]. The proportion of cytotoxic NK cells increased in the peripheral blood of TNBC patients after 12 weekly doses of chemotherapy, while CD4+ and CD8+ T cell fractions were maintained and those of PD-1+ CD4+ T cells were reduced [117]. Continued administration of chemotherapy, however, resulted in the depletion of NK cells to below baseline levels, whilst PD-1+ CD4+ T cells were significantly enhanced [117]. Notably, in alignment with the prognostic significance of infiltrating B cells at the primary tumour [11], peripheral B cell proportions were significantly more depleted in patients who did not survive beyond one year from the start of treatment compared to those who did [124]. Induction therapies capable of priming the TME may therefore augment the immunological landscape of other sites, including the peripheral blood. Data from on-treatment responses in LNs is lacking, presenting a knowledge gap and calling for further investigation in this field (Figure 4).

## 7. Anti-Tumour Effects of Radiation Therapies

Radiotherapy (RT) is often used as an adjuvant therapy following NACT or surgery and has shown immunomodulatory effects within the TME of breast carcinomas. The local effects of targeted irradiation on tumour cells are well documented and primarily consist of inducing irreparable DNA damage in highly proliferative cells leading to tumour cell death and cancer regression [125]. RT also affects leukocytes and can lead to lymphopenia in patients with breast cancer and other malignancies [126], potentially via systemic immunosuppression.

In pre-clinical models of breast cancer, tumour irradiation led to enhanced T cell-mediated cytotoxicity [127], increased levels of T cell-derived IFN-γ and TNF-α, and reduced infiltration of Tregs and MDCSs [128]. By contrast, the peripheral blood of breast cancer patients exhibits reduced numbers of lymphocytes and proportions of CD4+T cells and CD19+ B cells following standard RT [129] or intraoperative RT [130]. Similar to chemotherapy, reducing the dose and duration of RT can confer immunostimulatory effects in the peripheral blood. Whilst standard RT only induces transient increases in CD4+ T cell proportions and long-term CD19+ B cell depletion, a hypo-fractionated dose led to long-term expansion of both CD4+ T cell and CD19+ B cell proportions compared to levels observed immediately post treatment [129]. Therefore, hypo-fractionated lower-dose RT may be less harmful to CD19+ B cells in the long term whilst retaining the beneficial effect on T cells. In mice, the inclusion of the tumour-draining LNs within the irradiation field led to a significant increase in the amount of tumour infiltrating CD8+ T cells that expressed IFN-γ and TNF-α compared to tumour-targeting RT [131]. Nevertheless, irradiation of the tumour-draining LNs caused an overall reduction in the absolute count of tumour-infiltrating CD8+ T cells [131]. Similarly, elective nodal irradiation was shown to attenuate adaptive immune responses (when combined with CTLA-4 blockades) and adversely affected survival via reduced chemokine expression and immune infiltration [132]. These studies indicate that immunological cross-talk between the primary tumour, LNs and the peripheral blood also occurs in response to radiation and, in some cases, mirrors the observations made post-NACT. Further studies are required to optimise the use of nodal irradiation to prime immunologically ‘cold’ breast tumours without compromising ICB efficacy and patient survival.

In the clinical setting, RT is also known for its ability to induce ‘abscopal effects’, i.e., the occurrence of clinical responses to RT at distant, non-irradiated sites [133]. This phenomenon has been reported in breast cancer where non-irradiated metastatic lesions were seen to be reduced by >30% and improved clinical responses were associated with reduced baseline neutrophil to lymphocyte ratio [134,135], as observed following NACT [23,24]. However, positive abscopal effects following radiation are most frequently reported when RT is combined with ICBs, which exhibit increased efficacy at distant sites [136].

Pre-clinical studies provide robust support for the combination of RT with ICBs. In murine models of breast cancer, RT therapy improved local tumour control and prolonged survival by upregulating PD-L1 and PD-1 expression on tumour cells and T cells, respectively [128,137]. RT also potentiated the clinical efficacy of CTLA-4 blockades by enhancing T cell recruitment and infiltration of the tumour bed, stabilising NKG2D engagement with Rae-1 and increasing TCR clonality within the TME [137,138,139,140]. In breast cancer patients, RT failed to improve ORR to ICBs at both local and distant sites when administered in combination with PD-L1/PD-1 blockades [121,141], and only 18% of TNBC patients who were enrolled to receive RT plus pembrolizumab achieved durable complete responses. However, these patients demonstrated a 100% reduction in tumour burden at non-irradiated sites [142]. Additionally, pre-operative RT and pembrolizumab administered to TNBC patients in the neoadjuvant setting led to a partial (67%) or complete response which correlated with baseline TIL levels of >10% [143]. Whilst pre-clinical studies provide strong support for the combination of RT with ICBs in immunologically ‘cold’ tumours, additional studies are required to delineate and overcome the variable responses observed in breast cancer patients.

## 8. Implications of the Systemic Immune Response for Immunotherapy Efficacy

Immunotherapies are a promising option for breast cancer. Initially assessed as monotherapies, ICBs demonstrated moderate ORRs [144], in stark contrast to the high and durable responses observed in immunologically ‘hot’ tumours such as melanoma [145]. This became the rationale for combining ICBs with agents capable of converting poorly-immunogenic TMEs into immunologically dense ones. To date, only nab-paclitaxel has been approved in combination with atezolizumab or pembrolizumab, mAbs targeted at PD-L1 and PD-1, respectively, for the treatment of breast cancer in Europe [2,3,146]. These combinations build on the immunomodulatory potential of chemotherapy and have marginally improved treatment options for aggressive breast cancer subtypes. Immunotherapy combinations must be designed by leveraging strong biological rationale rather than feasibility alone, exemplified by the Impassion131 trial which combined atezolizumab with paclitaxel instead of nab-paclitaxel [147]. There was no improvement in OS for patients treated with combination therapy as compared to chemotherapy-only-treated patients. This may be explained by the necessary administration of immunosuppressive corticosteroids with paclitaxel.

The target location of immunotherapeutic stimulation has been thought to primarily be the TME itself rather than systemic immunity. Indeed, in the past, immunotherapy efficacies have been credited exclusively to the de-repression of T cell immune responses in the primary tumour. We have shown throughout this review that the systemic immune system of breast cancer patients is sensitive to the presence of tumour and anti-cancer therapies, putting forward that the peripheral blood and the tumour-draining LN are also likely to be affected by immunotherapy. Using mass cytometry, the immunotherapy-induced systemic immunological changes in a spontaneous mouse model of TNBC revealed large shifts in immune cell frequency and proliferation during treatment, which were sometimes mirrored between tissues [84]. Despite baseline intratumoural proliferation levels, TILs expanded, perhaps via infiltration from the periphery or the tumour-draining LN where immune proliferation was sustained during treatment [84]. In breast cancer patients, the immunological changes engendered by immunotherapy combinations have not been systematically assessed. Two recent studies provide single-cell atlases of the immune landscape of breast cancer patients before and after treatment with chemo-immunotherapy regimens [146]. Compared to non-responders, patients who responded to anti-PDL1 combination therapy presented with higher levels of pre-treatment peripheral T cell proliferation [148]. Moreover, their blood effector memory T cells exhibited higher levels of TCR clonality with a predictive population of intratumoural CXCL13-CD8+ T cells [148].

More than just being affected by immunotherapy, the peripheral blood and the tumour-draining LN are also reinvigorated by immunotherapeutic agents. As previously described, these sites often exhibit immunosuppression which will be reversed via immunotherapy such as ICBs. The targets for currently approved immunotherapy regimens in breast cancer are found within the tumour-draining LN. Indeed, the level of PD-1, the target of the FDA-approved pembrolizumab, is higher in sentinel LNs of breast cancer patients than non-sentinel LNs [48]. A novel immunotherapy platform using tumour-draining LN-targeted exosomes showed efficacy in reducing LN metastasis, suppressing tumour growth and prolonging survival in murine models of breast cancer [149]. Mice treated with the OX40-modulating exosome exhibited increased frequencies of effector T cells with enhanced IL-2 and IFN-γ signaling and reduced Treg induction in both their tumour-draining LNs and their primary tumours [149]. Such results are paving the way to directly targeting the immune landscape of the tumour-draining LN to improve immunotherapy efficacy in breast cancer.

Finally, recent preclinical evidence suggests that the systemic immune response, particularly in the tumour-draining LN, may be sufficient and required for immunotherapy response. Indeed, in an orthotopic model of breast cancer, the tumour-draining LN-targeted administration of immunotherapeutic agents (anti-CTLA-4 and anti-PD-1) is sufficient to suppress tumour growth to the same level as intratumoural administration and to a higher level than systemic administration [150]. This locoregional administration is achieved by ICB administration into the normal neighbouring skin tissue, which drains directly to the tumour-draining LN [150]. This has important implications for immunotherapy in the clinic since systemic administration of immunotherapeutic agents often leads to high toxicities. FTY720 is an immunomodulatory drug that abrogates T cell egress from lymphoid tissues. In multiple preclinical cancer models, including a spontaneous mouse model of TNBC, FTY720 treatment abolished tumour growth control and decreased TIL infiltration, suggesting that immunotherapy efficacy may depend on T cell infiltration from the tumour-draining LN [84,151,152,153]. Moreover, transferring T cells from the tumour-draining LN of immunotherapy-treated mice confers protection to naïve mice upon tumour challenge [84]. In a mouse model of colon cancer, tumour draining LN surgical resection after anti-PD-1 treatment decreased tumour control whilst tumour-draining LN resection before tumour induction completely abolished tumour control [152]. This suggests that, on top of being a source of TILs, the pre-treatment immune activity in the tumour-draining LN, i.e., antigen trafficking and lymphocyte priming, may play a pivotal role in the response to immunotherapy. Together, these studies present the tumour-draining LN as a key orchestrator of immunotherapy efficacy.

## 9. Clinical Application and Future Studies

The immune response to breast cancer is highly dynamic and truly systemic, yet immune characterisation of breast cancers rarely considers more than one time point or immune site. Considering both the marked responses to neoadjuvant therapy and the plasticity of systemic immune sites in breast cancer, one-dimensional immune profiling is no longer sufficient. The importance of immune dynamics has already been exemplified by Luen et al. who defined four immune response profiles, namely, ‘*persistent low*’ (L), ‘*fall in immunity*’ (F), ‘*immune induction*’ (II) and ‘*immune persistent* (IP) [154]. Unlike the traditional ‘high’ or ‘low’ TILs classifications, these profiles assess two time points, baseline and two weeks into neoadjuvant therapy, and describe the observed on-treatment changes to TILs. Across two cohorts of HER2+ breast cancer patients receiving neoadjuvant HER2 therapy, both II and IP patients had significantly better pCR rates compared to both L and F patients [154]. Importantly, these dynamic immune profiles consider the potential for TILs to increase or decrease significantly during neoadjuvant therapy, unlike traditional classifications that simply dichotomise disease progression. Considering the conflicting results of post-NACT TILs analyses [98,101], including a third time point after neoadjuvant therapy may also help to delineate these contradictions.

To evaluate all aspects of the immune response to breast cancer, the question for clinical practice now is ‘where’ in addition to ‘when’ should biopsies be taken. Considering the potent effect of systemic therapies, optimal biopsy collection would take place in both the neoadjuvant and on-treatment settings, as previously described. However, 20–30% of early-stage breast cancer patients are estimated to develop metastasis. For these patients, biomarkers are even more difficult to define as immune distinctions between sites are made even more clear and tissue heterogeneity is exacerbated in the adjuvant and residual disease settings. Notably, metastatic sites present a highly altered immune profile compared to the primary tumour and differ significantly depending on their physiological location [155]. Consistently, lung metastases are found to have high (median ~30%) TILs and immune cell PD-L1 positivity (mean ~70%), whilst skin and bone metastases have significantly lower TILs (median ~5%) and immune cell PD-L1 positivity (mean ~20%) [155,156,157], yet the prognostic significance of TILs in the advanced setting requires further investigation.

## 10. Conclusions

Despite advances in immunotherapies, their impact on OS for breast cancer patients continues to be limited by high levels of heterogeneity and poorly defined stratification markers for treatment groups. The primary tumour, the LN and the peripheral blood of breast cancer patients are distinct immunological sites which exhibit a breadth of anti-cancer immune functions. At the same time, these sites persistently communicate with one-another to mount and maintain immune responses and adapt to the ever-changing TME, but they are also sensitive to the systemic effects of anti-cancer therapies. Herein, future studies may seek to characterise systemic immune profiles of breast cancer patients in a ‘three-by-three’ manner, obtaining biopsies from all three of the primary tumour, lymph nodes and the peripheral blood at baseline, into neoadjuvant therapy and after treatment. In the advanced setting, metastatic TILs assessments may also be included to further delineate the progression of the systemic immune response to breast cancer and therapies. By considering all aspects in conjunction with each other, we can begin to delineate the highly dynamic and systemic labyrinth that constitutes the anti-tumour response to breast cancer.

## Figures and Tables

**Figure 1 cancers-14-04505-f001:**
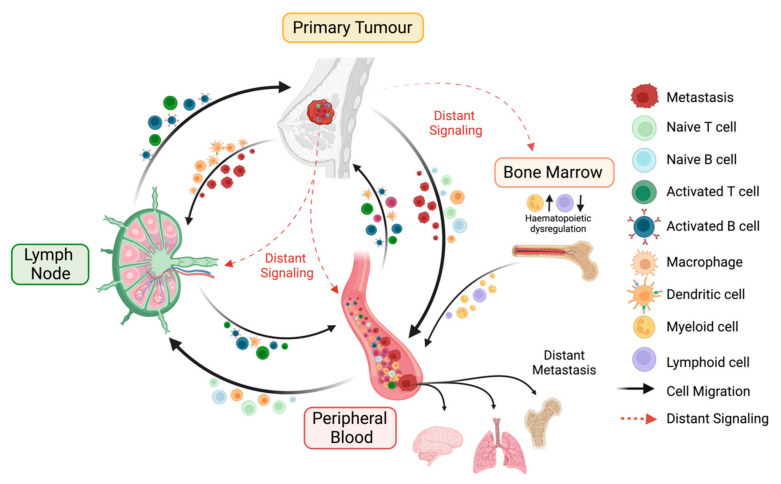
Key interactions of the immune macroenvironment. The cancer-immunity cycle is a coordinated process involving the primary tumour, the peripheral blood, and the tumour-draining lymph node (LN), with the potential to generate effective anti-tumour immune responses. The primary tumour can either interact directly with the peripheral blood and the LN due to the dissemination of cancer cells or indirectly via the secretion of distant immunosuppressive mediators. The presence of a tumour induces hematopoietic dysregulation via increased abundances of hematopoietic stem cells (HSCs) and granulocyte monocyte progenitors (GMPs) in the bone marrow, followed by peripheral blood alterations where myeloid and lymphoid cells are increased and decreased, respectively. Tumour antigens secreted from the primary tumour can travel to the two other sites. Antigen presenting cells (APCs), e.g., macrophages and dendritic cells (DCs), which have encountered these antigens, either at the tumour lesion or in the periphery, are trafficked to the LN, where they prime and activate naïve lymphocytes. These primed effector immune cells then infiltrate the tumour to carry out effector functions or enter the circulation as memory subsets.

**Figure 2 cancers-14-04505-f002:**
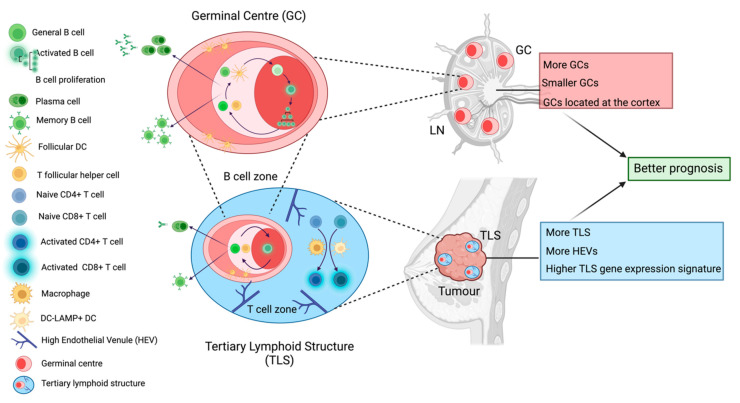
Tertiary lymphoid structures and germinal centres within the lymph node. Both tertiary lymphoid structures (TLSs) and germinal centres hold cellular and functional similarities which are associated with better prognosis in breast cancer patients. Germinal centres are dynamic immune structures within LNs which lead to the generation of affinity matured antibody-producing cells and memory B cells via strong interactions between activated B cells and T follicular helper (Tfh) cells as well as follicular dendritic cells (FDCs). The germinal centre is organised into two zones: the light zone where these intercellular interactions occur and the dark zone where selected B cells proliferate. We have found that LN-positive TNBC patients who present with more, smaller and cortically located germinal centres in their cancer-free LNs exhibit longer time to distant metastasis. Remarkably, mature TLSs within the primary tumour contain a germinal centre-like aggregate encircled by a T cell zone where macrophages and dendritic cells activate naïve T cells. TLSs are characterised by the presence of high endothelial venules (HEV). This germinal centre also leads to the generation of antibody-producing and memory B cells. TNBC patients who exhibit more histologically detected TLSs, more HEVs, and whose tumours express higher levels of TLS gene expression signatures, have better prognosis.

**Figure 3 cancers-14-04505-f003:**
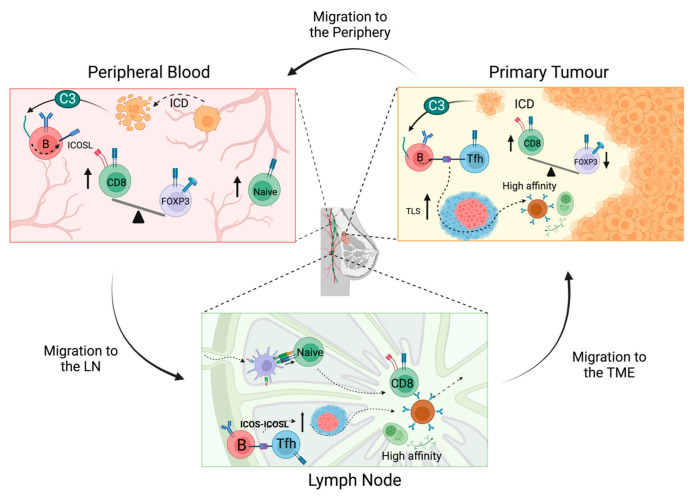
Immunostimulatory effects of chemotherapy across the immune macroenvironment. In the TME, chemotherapy results in an expansion of ICOSL+ B cells, increased TLSs and increased CD8:Treg ratios. ICOSL+ B cells accumulate in TLSs, where they may promote B cell survival in a germinal centre-like response to generate high affinity memory B and plasma cells. Chemotherapy induces immunological cell death in metastatic tumour cells migrating through the periphery and may lead to the upregulation of ICOSL+ B cells. Chemotherapy increases the CD8:Treg cell ratio in the peripheral blood and the proportion of naïve CD4+ and CD8+ T cells. Increased levels of naïve T cells and ICOSL+ B cells may migrate from the periphery to the LN, leading to enhanced activation of effector T cells and promotion of B cell survival and memory B/plasma cells. Activated effector T cells and high affinity B cells from the LN then egress back to the TME where they mediate enhanced anti-tumour activity.

**Figure 4 cancers-14-04505-f004:**
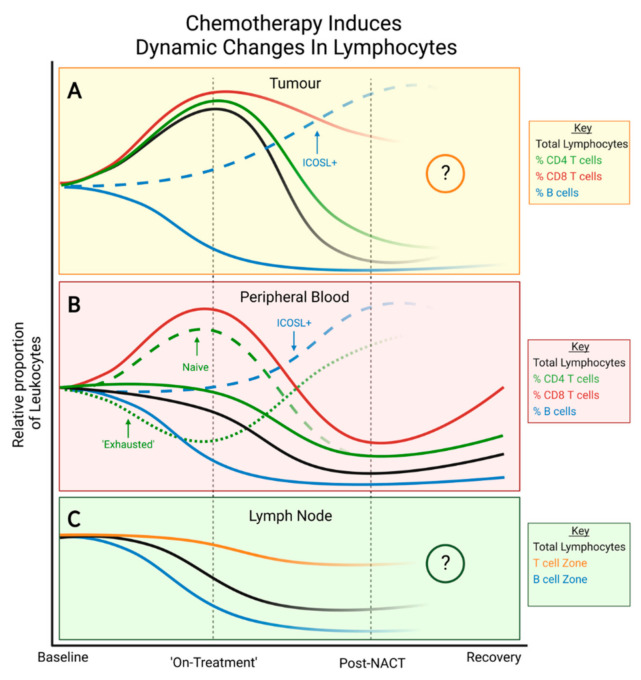
Immunomodulatory effects of chemotherapy during treatment. (**A**) Within an immune-infiltrated TME, proportions of CD4+ and CD8+ T cells increase initially before falling after NACT. Post-NACT, CD4+ T cells, B cells and regulatory T cells are depleted compared to baseline levels. (**B**) In peripheral blood, proportions of NK cells, naïve CD4+ T cells and naïve CD8+ T cells are initially increased whilst PD-1+ CD4+ T cells with an ‘exhausted’ phenotype are depleted compared to baseline. Post-NACT, all subtypes are depleted compared to baseline levels except ‘exhausted’ CD4+ T cells which are increased compared to both baseline and on-treatment values. Immune cell subtypes are variably depleted with B cells being the most affected, followed by both CD4+ subtypes. Both CD8+ subtypes and NK cells are less affected. All subtypes recover to a degree, but B cells and CD4+ T cells remain low for longer. To date, no studies have evaluated the long-term effects of NACT on PD-1+ CD4+ T cells. (**C**) In cancer-free LNs, post-NACT, total lymphocytes are depleted compared to baseline levels with B cell zones being more diminished than T cell zones. Studies are warranted to evaluate the on-treatment or long-term effects of NAT on lymphocytes within the cancer-free LN.

**Table 1 cancers-14-04505-t001:** Studies characterising the immune profiles of the tumour microenvironment, the peripheral blood and the lymph nodes.

IMMUNE SITE	PATIENT COHORT/ PRECLINICAL MODEL	IMMUNE COMPONENT	IMMUNE ACTIVATION OR SUPPRESSION?	MAJOR OBSERVATION	REF
**Tumour Microenvironment (TME)**					
TME	TNBC; ER+	Spatial immune phenotype	Both	Stratification of breast TME using spatial immune phenotypes.	[12,13,14]
TME	TNBC	Spatial immune phenotype	IA	Inflamed spatial immune phenotype associated with improved MDS, DFS and OS.	[13]
TME	Not stratified	TLS	IA	12-chemokine TLS signature predicts improved survival.	[15]
TME	TNBC	TLS	IA	Histologically-detected TLS are prognostic for OS.	[16]
TME	Luminal A, Luminal B, HER2+, TNBC	TLS	IA	TLS associated with improved DFS and OS.	[17]
TME	HER2+, TNBC	TIL-B	IA	TIL-B are activated, secrete cytokines and respond to ex vivo BCR stimulation.	[18]
TME	TNBC	TIL-B	IA	IgG isotype switched TIL-B are associated with favourable prognosis.	[19]
TME	TNBC	γδ-T cells	IA	γδ-T cells possess a tumour-rejecting phenotype and their presence in tumours is associated with improved PFS and OS.	[20]
TME	Murine model of breast cancer	γδ-T cells	IS	IL-17-producing γδ-T cells dampen T cell responses and block DC maturation.	[21,22]
**Peripheral Blood (PB)**					
PB	Luminal A, Luminal B, HER2-enriched, TNBC	NLR	IS	High NLR is associated with worst prognosis.	[23,24]
PB	Luminal A, Luminal B, HER2-enriched, TNBC	LMR	IA	High LMR is predictive for improved response to treatment.	[23]
PB	Luminal A, Luminal B, HER2-enriched, TNBC	Circulating lymphocytes	IA	High circulating lymphocyte counts associated with better outcome.	[25]
PB-TME	Luminal A, Luminal B, Normal-like, Basal-like, HER2-enriched	PB RNA profiles	Both	Peripheral inflammation is associated with intratumoural immune activation.	[26]
PB	HR+, HR+HER2+, HR-HER2+, TNBC	T cells	IS	Peripheral T cells exhibit impaired cytokine secretion, responsiveness and reduced TCR signaling.	[27]
PB	Early BC	CD8+ T cells	IS	Circulating CD8+ T cells display senescent and exhausted phenotypes.	[28]
PB-TME	TNBC	CD4+ and CD8+ T cells	IS	T cell exhaustion in the PB and in the TME are correlated.	[29]
PB	ER+, PR+, HER2+	NK cells	IS	Circulating NK cells express less activation receptors and are less cytotoxic.	[30]
PB	Luminal A, Luminal B, HER2+, TNBC	Dendritic cells	IS	Circulating DCs are less mature and express lower activation marker levels.	[31]
PB	ER+, PR+, HER2+, TNBC	Dendritic cells	IS	Circulating DCs produce less TNF-α	[32]
PB	Murine model of breast cancer	Dendritic cells	IS	Tumour G-CSF inhibits PB DC maturation.	[33]
PB	Luminal, HER2+, TNBC	Tregs	IS	BC patients have more circulating Tregs which are related to RFS.	[34]
PB	Luminal, HER2+, TNBC	Monocytes	IS	Peripheral monocytes are less responsive to interferons.	[35]
PB	Pan-Cancer	MDSC	IS	MDSCs are increased in the PB of breast cancer patients and associated with disease stage.	[36]
PB	ER+, PR+, HER2+, HER2-enriched	MDSC	IS	Peripheral MDScs inhibit T cell proliferation and exhibit immunosuppressive markers.	[37]
PB	Murine model of breast cancer	Neutrophils	IS	G-CSF induced circulating neutrophil expansion inhibits T cell function and induces metastasis.	[38]
PB	Luminal A, Luminal B, HER2-enriched, TNBC	PLR	IS	BC patients with low platelet to lymphocyte ratio have significantly higher cPR rates, independent of breast cancer subtypes.	[39]
PB	TNBC	PLR	N/A	PLR is associated with a favourable response to NACT in TNBC patients.	[40]
PB	Luminal A, Luminal B, HER2+, TNBC	PLR	IS	PLR is an independent prognostic factor for RFS and shorter OS.	[41]
PB	Luminal A, Luminal B, HER2-enriched, TNBC	SII	N/A	SII is predictive for OS in BC.	[42,43]
**Lymph Node (LN)**					
MLN	Not stratified	T cells	IS	CD4/Cd8 ratio is decreased compared to non-MLN.	[44]
MLN	HR+, HER2+, TNBC	MDSC	IS	MSCS are expanded in the MLN.	[45]
MLN	Luminal	Tregs	IS	MLN Tregs are increased and express higher levels of inhibitory molecules.	[46]
MLN	Not stratified; Luminal A, Luminal B, HER2+, TNBC	Immune markers	IS	MLNs exhibit increased levels of IL-10, FOXP3, CTLA-4 and PD-1.	[47,48]
MLN	HR+, HER2+, TNBC	T cells	IS	MLN T cells exhibit a more exhausted phenotype and secrete less pro-inflammatory cytokines.	[45]
MLN	HR+, HER2+, TNBC	Dendritic cells	IS	MLN DCs are less activated and respond less to ex vivo stimulation.	[45]
MLN	Luminal	Macrophages	IS	MLNs have more TAMs, which secrete IDO.	[49]
MLN, TME	LN+ (TNBC)	Germinal centre, TLS	IA	Tumour TLS presence is associated with MLN GCs.	[50]
CFLN	Not stratified	T cells	IS	CFLN harbour less CD4+ T cells and less CD8+ T cells.	[44]
CFLN	Not stratified	Dendritic cells	IS	CFLN harbour DCs with poor antigen presentation characteristics.	[51]
CFLN	HR+, HER2+, TNBC	T cells	IA	CFLN harbour more CD3+ T cells than MLN.	[52]
CFLN	Not stratified	Dendritic cells	IA	CFLN harbour more DCs which are more mature than in the MLN.	[44,51]
CFLN, MLN, PB, TME	Luminal	T cells	IA	Matched investigation of T cell phenotype and tumouricidal activity.	[53]
CFLN	LN+ (HR+, HER2+, TNBC; TNBC)	Germinal centre	IA	GCs in CFLNs are associated with longer DMFS.	[50,54]
CFLN, TME	LN+ TNBC	Germinal centre, TILs	IA	High TIL levels are associated with CFLN GCs.	[50]
CFLN, TME	LN- TNBC	Germinal Centre, TILS	IA	High TILs patients harbour more and bigger GCs in their CFLNs	[55]
CFLN	TNBC	LN size	IA	Enlarged LN without nodal involvement are associated with longer survival.	[56]
CFLN, MLN	Not stratified	T cells and B cells	IA	Increased sentinel LN T and B cells are associated with longer DFS.	[52]
CFLN, MLN	LN+ patients	T cells and DCs	IA	Increased axillary CD4+ T cells and DCs are associated with longer DFS.	[44]
CFLN	LN- TNBC	Immune checkpoint molecules	IS	High LN immune checkpoint molecule expression is associated with low TIL Levels.	[57]

TME, Tumour Microenvironment; PB, Peripheral Blood; LN, Lymph Node; MLN, Metastatic Lymph Node; CFLN, Cancer-Free Lymph Node; HR, Hormone Receptor (Estrogen and Progesterone); ER, Estrogen Receptor; PR, Progesterone Receptor; HER2, Human Epidermal-Growth-Factor Receptor 2; TNBC, Triple-Negative-Breast-Cancer; IA, Immune Activation; IS, Immune Suppression; NLR, Neutrophil-to-Lymphocyte Ratio; LMR, Lymphocyte-to-Monocyte Ratio; PLR, Platelet-to-Lymphocyte Ratio; GCs, Germinal Centres.

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
