# Peer review of "Leveraging the Dynamic Immune Environment Triad in Patients with Breast Cancer: Tumour, Lymph Node, and Peripheral Blood"

_cancers, 2022, doi:10.3390/cancers14184505_

Round 1
Reviewer 1 Report
This is a useful, comprehensive, and clinically relevant review of the literature on the interplay of three components of the immune environment in breast cancer patients: tumor, lymph nodes, and peripheral blood. Overall, the selection of the cited studies is careful, the presentation of the data is logical, and the Figures are informative.
The high quality of the submitted work clearly predestines it for publication. My only concern is that despite the title, the review reduces the immunological environment of the "peripheral blood" mainly to the leukocytes, while, e.g., platelets knowingly play a crucial role in tumor invasion and metastasis, also via immunological pathways.
In addition, the rarity of the data is repeatedly emphasized, although fortunately several directly or indirectly related publications have appeared in recent years that are not included in the submitted review.Some examples are: Urueña et al. 2022 (PMID: 35562400), Ahn et al. 2020 (PMID: 32401825), Wang et al. 2022 (PMID: 35664009); Noël et al. 2021 (PMID: 34411002); Grigoriadis et al. 2018 (PMID: 29416876), Chatterjee 2018 (PMID: 30458865), Kim et al. 2021 (PMID: 33157515).
I hope that my minor comments can help the authors to improve their work from "very good" to "outstanding".
Author Response
Point 1: This is a useful, comprehensive, and clinically relevant review of the literature on the interplay of three components of the immune environment in breast cancer patients: tumor, lymph nodes, and peripheral blood. Overall, the selection of the cited studies is careful, the presentation of the data is logical, and the Figures are informative.
Response 1: No response necessary.
Point 2: The high quality of the submitted work clearly predestines it for publication. My only concern is that despite the title, the review reduces the immunological environment of the "peripheral blood" mainly to the leukocytes, while, e.g., platelets knowingly play a crucial role in tumor invasion and metastasis, also via immunological pathways.
Response 2: We agree that the role of platelets are important and have therefore included a paragraph describing their role in immune responses and cancer progression in the neoadjuvant and adjuvant settings. Please see lines 243-260 and 432-436.
Point 3: In addition, the rarity of the data is repeatedly emphasized, although fortunately several directly or indirectly related publications have appeared in recent years that are not included in the submitted review.Some examples are: Urueña et al. 2022 (PMID: 35562400), Ahn et al. 2020 (PMID: 32401825), Wang et al. 2022 (PMID: 35664009); Noël et al. 2021 (PMID: 34411002); Grigoriadis et al. 2018 (PMID: 29416876), Chatterjee 2018 (PMID: 30458865), Kim et al. 2021 (PMID: 33157515).
Response 3: We have included all citations listed by the reviewer, they can be found by the following citation numbers: 22, 23, 108, 19, 91, 83, 103 (given in the order as written in Point 3 by the reviewer).
Reviewer 2 Report
it would be helpfull ti include a table with all the reviews- studies that examine the immune profiles, excellent work , well established
Author Response
Point 1: it would be helpfull ti include a table with all the reviews- studies that examine the immune profiles, excellent work , well established
Response 1: We agree a table would be beneficial and have therefore included the requested information in 'Table 1' which can be found at the bottom of the manuscript on page 19, before the references.